# Transcriptomic and Metabolomic Studies Reveal That Toll-like Receptor 2 Has a Role in Glucose-Related Metabolism in Unchallenged Zebrafish Larvae (*Danio rerio*)

**DOI:** 10.3390/biology12020323

**Published:** 2023-02-16

**Authors:** Wanbin Hu, Li Liu, Gabriel Forn-Cuní, Yi Ding, Alia Alia, Herman P. Spaink

**Affiliations:** 1Institute of Biology Leiden, Animal Science and Health, Leiden University, Einsteinweg 55, 2333 CC Leiden, The Netherlands; 2Institute for Medical Physics and Biophysics, University of Leipzig, 04107 Leipzig, Germany; 3Leiden Institute of Chemistry, Leiden University, 2333 CC Leiden, The Netherlands

**Keywords:** transcriptomics, metabolomics, glycolysis, glucose metabolism, *tlr2*, zebrafish

## Abstract

**Simple Summary:**

Toll-like receptor 2 (TLR2) has been demonstrated to participate in the progression of some metabolic disorders due to its role as a pro-inflammatory trigger. However, whether TLR2 plays a role in mediating metabolism under an unchallenged condition is still unknown. Therefore, we utilized zebrafish larvae as an in vivo model to investigate the metabolic control functions of TLR2 through transcriptomic and metabolomic approaches at a whole-organism level. We found that the concentration of glucose, lactate, succinate, and malate is higher in a *tlr2* mutant which is associated with lower expression of genes involved in the glycolysis and gluconeogenesis pathways. These results demonstrate that *tlr2* plays a role in controlling glucose metabolism homeostasis.

**Abstract:**

Toll-like receptors (TLRs) have been implicated in the regulation of various metabolism pathways, in addition to their function in innate immunity. Here, we investigate the metabolic function of TLR2 in a larval zebrafish system. We studied larvae from a *tlr2* mutant and the wild type sibling controls in an unchallenged normal developmental condition using transcriptomic and metabolomic analyses methods. RNAseq was used to evaluate transcriptomic differences between the *tlr2* mutant and wild-type control zebrafish larvae and found a signature set of 149 genes to be significantly altered in gene expression. The expression level of several genes was confirmed by qPCR analyses. Gene set enrichment analysis (GSEA) revealed differential enrichment of genes between the two genotypes related to valine, leucine, and isoleucine degradation and glycolysis and gluconeogenesis. Using ^1^H nuclear magnetic resonance (NMR) metabolomics, we found that glucose and various metabolites related with glucose metabolism were present at higher levels in the *tlr2* mutant. Furthermore, we confirmed that the glucose level is higher in *tlr2* mutants by using a fluorometric assay. Therefore, we have shown that TLR2, in addition to its function in immunity, has a function in controlling metabolism during vertebrate development. The functions are associated with transcriptional regulation of various enzymes involved in glucose metabolism that could explain the different levels of glucose, lactate, succinate, and malate in larvae of a *tlr2* mutant.

## 1. Introduction

Toll-like receptor 2 (TLR2) is one of the most well-characterized pattern recognition receptors (PRRs) and is ubiquitously expressed on the different types of immune cells [1,2]. TLR2 forms a heterodimer with TLR1 or 6 to recognize pathogen-associated molecular patterns (PAMPs) of invading microbial pathogens [3,4,5]. In addition, TLR2 can also sense damage-associated molecular patterns (DAMPs) released by damaged tissue, lysing cells, or disrupted extracellular matrix [6,7]. As soon as TLR2 or its heterodimers bind with PAMPs or DAMPs, adaptor proteins MYD88 and TIRAP (also called MAL) can be recruited to subsequently activate the intracellular NF-κB signaling pathway to induce cytokines and chemokines secretion [2,3,4]. Therefore, the focus of studies of TLR2 has been mainly linked to its pro- or anti-inflammatory functions, with little attention to proposed roles in physiological processes such as glucose metabolism [2].

A wide range of molecular and cellular processes, including transcription, translation, and biosynthesis, are dependent on glucose metabolism [8,9,10,11,12]. Macrophages and dendritic cells upon LPS infection show increased aerobic glycolysis, which is crucial for regulating the activation of these cells [13]. In tuberculosis (TB) studies, aerobic glycolysis metabolism is increased, and the activity of the citric acid cycle is decreased as soon as the macrophages are activated and changed into a pro-inflammatory M1 phenotype upon *Mycobacterium tuberculosis* infection [14]. Notably, TLR2 has been demonstrated to be involved in this process [15]. In addition, monocarboxylate transporter 4 (MCT4), which exports cellular lactate, is upregulated by TLR2 in macrophages [16]. It has been demonstrated that hypoxia-inducible factor 1 alpha (HIF1A) can be induced through LPS stimulated- TLR4 signaling, which ultimately regulates the expression of metabolic genes such as the phosphoglycerate kinase 1 (*Pgk1*) and glucose transporter 1 (*Glut1*) genes [17,18]. However, the role of TLR2 in glucose metabolism and the regulation of glucose transporters is still unclear.

Zebrafish larvae have been demonstrated to be a good model organism for investigating metabolism [19,20,21]. Because zebrafish can produce a large number of offspring, it makes them suitable for use in high-throughput screening methods [22,23,24]. In addition, the small body size of zebrafish larvae makes them suitable for omics studies of the whole organism. The TLR signaling pathway is highly conserved between mammals and zebrafish [25]. We previously demonstrated that a synthetic triacylated lipopeptide Pam3CSK4, which is a mammalian TLR2 ligand that can specifically activate the zebrafish Tlr2 pathway, induced the upregulation of *fosl1a* and *cebpb* [26]. Furthermore, the function of *tlr2* in innate immune responses has been characterized in zebrafish [27,28,29]. In our previous paper, we found that *tlr2* regulates leukocyte migration upon tissue wounding and mycobacterial infection [28,29]. In addition, we also found that a mutant in *tlr2* showed a large difference in the transcription level of genes involved in glycolysis as compared to heterozygote control zebrafish larvae under unchallenged condition [27]. This is unexpected since the function of *tlr2* has only been investigated in inflammation. Therefore, we further investigated the function of the *tlr2* in glucose metabolism in this study using further outbred animals and homozygous controls.

We used RNA deep sequencing (RNAseq) to identify a set of genes involved in metabolism that is significantly altered in gene expression in *tlr2* mutant larvae compared to the wild type controls under an unchallenged condition. The ^1^H nuclear magnetic resonance (NMR) metabolomics was used to study metabolic changes between these homozygous genotypes. The NMR analysis identified many differences in small metabolites that are important for energy homeostasis. In addition, we quantified glucose levels using a fluorometric method. We discuss how changes in gene expression could explain the observed difference in metabolite levels between *tlr2* mutant and wild type larvae.

## 2. Materials and Methods

### 2.1. Zebrafish Maintenance and Samples Collection

All zebrafish were handled in compliance with the local animal welfare regulations and maintained according to standard protocols (zfin.org). The culture of zebrafish with mutations in immune genes was approved by the local animal welfare committee (DEC) of the University of Leiden (protocol 14,198). All protocols adhered to the international guidelines specified by the EU Animal Protection Directive 2010/63/EU.

The *tlr2^sa19423^* (further referred as *tlr2^−/−^* or *tlr2* mutant) line (ENU-mutagenized) was obtained from the Sanger Institute Zebrafish Mutation Resource (Hinxton, Cambridge, UK) and shipped by the zebrafish resource Center of the Karlsruche Insititue of Technology. *Tlr2* mutant and *tlr2* wild type fish were screened and raised as previous publications [27,28]. Homozygote carriers of the mutations were all outcrossed more than five times against wild type (ABTL strain). Homozygote mutants and their wild type control were used in this study. Three pairs of 2-year-old *tlr2^+/+^* or *tlr2^−/−^* zebrafish were bred by a single cross. Subsequently their offspring were collected at 5 days post fertilization (dpf). To link the results from RNAseq sequencing and NMR spectroscopy, the *tlr2* zebrafish larvae from one single cross were divided into two tubes for preparing the samples of RNAseq and NMR spectroscopy (Figure 1A).

### 2.2. RNA Isolation, cDNA Synthesis and qPCR

To perform the deep sequencing and qPCR, the extraction of total RNA from 5 dpf zebrafish larvae in *tlr2* wild type and *tlr2* mutant groups using TRIzol Reagent (Life Technologies, Carlsbad, CA, USA) according to the manufacturer’s instructions. To remove the DNA contamination, DNase treatment was conducted by using the kit (Thermo Fisher Scientific, Waltham, United States). RNA integrity and concentration detection and the synthesis of cDNA were conducted according to the methods described by Yang et al. [26]. Subsequently, we performed qPCR on a CFX96TM Touch Real-Time PCR Detection (Bio-Rad Laboratories, Inc., Hercules, CA, USA to quantify the gene expression profiles in the glucose metabolism pathway from *tlr2* wild type and mutant groups. qPCR of targeted genes expression was normalized against the expression of *ppial* as a reference gene [30]. The primer sequences used in this study are shown in Appendix A. qPCR reaction procedure was performed using the following protocol: 95 °C 3 min, 40 cycles real time of 95 °C 15 s, 68 °C 30 s and 72 °C 30 s, and final melting curve of 95 °C 1 min and 55 °C 10 s. The qPCR assay was biologically repeated for more than three times, and the relative expression level was determined by the comparative 2^−ΔΔCt^ method [31].

### 2.3. RNAseq Processing and Analysis

RNA sequencing of unchallenged *tlr2* mutant and wild type zebrafish larvae were conducted by GenomeScan B.V. (Leiden, The Netherlands) as previously described [32]. Sequencing data of three biological replicates for the *tlr2* mutant and wild type control were aligned and mapped to the zebrafish genome GRCz11 using Salmon v1.2.1 [33]. Differential gene expression was analyzed by using DESeq2 v1.24.0 [34]. Statistical significance was determined by an *S* value of ≤ 0.005 by utilizing apeglm [35]. Recently, *S* values, which are aggregate statistics, have been offered as a replacement to adjusted *p* values and false-discovery rates (FDRs) to calculate the probability of obtaining the sign of an effect wrong in biological contexts [36,37]. Gene Ontology (GO) term enrichment was performed in DAVID Bioinformatics Resources 6.8 (https://david.Ncifcrf.Gov/ accessed on 10 February 2023), while gene set enrichment analysis (GSEA) using the C2 “Curated Gene Sets” collections from Molecular Signatures Database (MsigDB) was conducted as described before [37,38].

### 2.4. NMR Sample Preparation

For the purpose of NMR spectroscopy, the method used to extract metabolites from zebrafish larvae in wild type and mutant groups was based on our previous research methods [32,39,40]. Briefly, each sample contained 120 zebrafish larvae at 5 dpf in a mixture of methanol: water (1:1, *v/v*) with 1 mL of chloroform. The mixture was processed by sonication for 15 min and then centrifuged at 5000 rpm for 5 min. After centrifugation, two layers were formed, and the methanol and water from the upper layer was collected. The dried methanol: water layer containing metabolites was dissolved in 1 mL of 100 mM deuterated phosphate (KD2PO4, pH: 7.0) containing 0.02% trimethylsilyl propionate (TSP) as an internal standard and followed by filtration with a Millipore filter (Millex-HV0.45-lmFilterUnit). Metabolites in zebrafish larvae were measured using a Bruker DMX 600 MHz NMR spectrometer at 4 °C equipped with a 5 mm inverse triple high-resolution probe with an actively shielded gradient coil. The ^1^H NMR spectra were accumulated with 65,000 data points, a relaxation delay of 2 s, a scan width of 12.4 kHz and 128 scans to obtain a satisfactory signal-to-noise ratio.

### 2.5. NMR Measurement and Analysis

NMR analysis was performed based on previous research [32,40,41,42]. First, one-dimensional (1-D) ^1^H NMR spectra obtained from both in the wild type and mutant groups were corrected for baseline and phase shifts using MestReNova software version 11.0 (Mestrelab Research S.L., Santiago de Compostela, Spain). The spectra were then subdivided into buckets of 0.04 ppm in the range of 0 to 10.00 ppm, and the spectra from chemical shift 0.80 to 4.30 ppm were assigned to specific metabolites. To remove the water peak, the region of 4.30–6.00 ppm was excluded from the analysis. The data matrix obtained was exported into Microsoft office excel (Microsoft Corporation, Redmond, DC, United States). These data were then simultaneously imported into MetaboAnalyst 4.0 for Partial least squares-discriminant analysis (PLS-DA) and heatmap analysis. A correlation coefficient of *p* < 0.05 was considered statistically significant. Metabolite quantification was performed using Chenomx NMR Suite 8.3, which allowed both qualitative and quantitative analysis of NMR spectra by fitting spectral features from the HMDB database to the spectra.

### 2.6. Glucose Measurement

The measurement of glucose content from both the wild type and mutant groups was performed using PicoProbeTM Glucose Fluorometric Assay Kit (BioVision, Milpitas, CA, USA) following the manufacturer’s instructions. In brief, five pooled larvae (5 dpf) from the same group were sampled as a specimen and homogenized for 3 min in 100 µL glucose assay buffer. The supernatant was then collected by centrifugation at 12,300 rpm for 30–40 min. Subsequently, we added a 10 µL sample into a 96-well plate and adjusted the volume to 45 µL with glucose assay buffer. Then, we added 5 µL Reaction Mix (2.5 µL glucose assay buffer, 0.5 µL PicoProbeTM, 1 µL glucose enzyme mix, and 1 µL glucose substrate mix) to each well and mixed well. To obtain the standard curve of glucose concentration, glucose standard with known concentration was diluted to obtain a concentration gradient; then, similarly 5 µL of Reaction Mix was added to each well, finally adjusting the volume to 50 µL with glucose assay buffer. The 96-well plate was incubated for 30 min at 37 °C and kept in the dark for the reaction. Finally, the fluorescence at Ex/Em = 540/590 nm was measured in a PHERAstar^®^ FSX (BMG LABTECH, Ortenberg, Germany) microplate reader. A standard curve of glucose concentration was made, and the glucose concentration of each sample was calculated from this.

### 2.7. Statistical Analyses

Graphpad Prism software (version 8.1.1; Graphpad Software, San Diego, CA, USA) was utilized for statistical analysis in Figures 3, 5 and 6. All experiment data are shown as mean ± SD, and unpaired two-tailed *t*-test was applied. Significance was established at *p* < 0.05, and the other significance levels are indicated as * *p* < 0.05; ** *p* < 0.01; *** *p* < 0.001.

## 3. Results

### 3.1. Transcriptomic Profiling of Tlr2 Mutant Zebrafish

In our previous study, we found several genes involved in glucose metabolism are differentially expressed in the transcriptome of a *tlr2* homozygote mutant versus *tlr2* heterozygote mutant zebrafish larvae [27]. To obtain more insights on the function of *tlr2* in glucose homeostasis, we performed deep RNA sequencing of *tlr2* mutant zebrafish larvae using further outbred animals and homozygous controls under an unchallenged condition (Figure 1).

To compare the difference between the *tlr2* mutant zebrafish larvae and wild type controls, we firstly summarized the number of DEGs in *tlr2^−/−^* versus *tlr2^+/+^* zebrafish larvae groups (Figure 1B). In total, 149 genes were differentially regulated, including 89 upregulated genes and 60 downregulated genes (Figure 1B). The details of these genes are shown in Appendix A. To investigate the identified DEGs, GO analysis was performed by using the DAVID bioinformatics program (http://david.abcc.ncifcrf.gov/ 10 February 2023) (Figure 1C).

### 3.2. GSEA Analysis of Tlr2 Mutant Zebrafish

We performed GSEA [43] to identify the classes of differentially enriched genes associated with *tlr2^+/+^* or *tlr2^−/−^*. GSEA predicted that 1136 gene sets were significantly enriched at FDR < 0.25 in *tlr2^+/+^*, while 613 gene sets were significantly enriched at FDR < 0.25 in *tlr2^−/−^* zebrafish larvae by using the C2 collection for curated gene sets. In Appendix A, the pathways found to be the top 10 significantly enriched in *tlr2^+/+^* and *tlr2^−/−^* are shown. Five metabolic associated pathways were found in the top 10 most significantly enriched pathways in *tlr2^+/+^*, including valine leucine and isoleucine degradation, glycolysis and gluconeogenesis, tryptophan metabolism, fatty acid metabolism, and pyruvate metabolism (Figure 2A–E). Furthermore, the selenoamino acid metabolism gene set was found in the top 10 most enriched pathways in the *tlr2* mutant group (Figure 2F). Taken together, the GSEA analysis results indicate that Tlr2 plays a role in controlling metabolic during homeostasis.

In concordance with GSEA results, we found that *gpib*, *pfkma*, and *pck2*, which are associated with the glycolysis and gluconeogenesis pathways, were significantly downregulated in the RNAseq analysis (Table 1). To validate the gene expression data from RNAseq, qPCR was conducted from 5 dpf *tlr2* zebrafish larvae, which confirmed these results (Figure 3).

### 3.3. System Metabolomics Analysis of Tlr2 Mutant Zebrafish

The observed transcriptional effects of the *tlr2* mutation are mainly related to the metabolic signaling pathways (Appendix A). Therefore, we decided to utilize a ^1^H NMR spectroscopy to study differences in metabolites levels between *tlr2* wild type and *tlr2* mutant zebrafish larvae.

Metabolic profiles of extracted zebrafish larvae at 5 dpf were obtained by one-dimensional (1-D) ^1^H NMR. Representative 1-D ^1^H NMR spectra from *tlr2* wild type and mutant groups are shown in Figure 4A. Chemical shifts of 1-D ^1^H NMR in two groups were assigned according to the chemical shifts of reference metabolites from the Chenomx 600 MHz library (version 11). By multivariate analysis with Partial least squares-discriminant analysis (PLS-DA) modeling, the ^1^H NMR spectra of *tlr2* wild type and mutant groups were investigated to probe if these two experimental groups can be well discriminated. As shown in Figure 4B, the first two principal components of the PLS-DA scores plot explained 94.8% of the total variance. Clustering of the *tlr2* wild type and mutant larvae groups can be observed in the PLS-DA1 versus PLS-DA2 score plots, suggesting metabolic alterations resulting from *tlr2* deficiency in zebrafish larvae. Heatmap analysis was performed on 57 metabolites detected in both groups (Figure 4C). Two of these metabolites were significantly increased in the *tlr2* wild type group, and 27 were significantly increased in the *tlr2* mutant group (*p* < 0.05). The details of these significantly altered metabolites are shown in Appendix A.

We analyzed metabolites that are linked to the pathways highlighted by the RNAseq analyses. In the valine, leucine, and isoleucine degradation pathway, five metabolites, leucine, valine, isoleucine, acetoacetate, and methylmalonate, were detected using NMR analysis, with the contents of leucine and valine being significantly higher in the mutants (Figure 5A). In the glycolysis and gluconeogenesis pathway, the contents of glucose, lactate, acetate, formate, malate, and succinate showed increase trends in the mutant group compared with the wild type larvae, with the exception of pyruvate and fructose (Figure 5B). In the tryptophan metabolism pathway, the content of melatonin was significantly reduced in the mutants, while the contents of indole-3-acetate and kynurenine were significantly increased in the mutants compared with the wild type larvae (Figure 5C). For the fatty acid metabolism pathway, glycerate content was significantly reduced in the mutant larvae, while the contents of taurine, glycine, and ethanolamine were significantly increased in the mutant compared with the wild type larvae (Figure 5D).

### 3.4. Glucose Levels and Transcription of Glycolytic Enzymes in tlr2 Mutant Zebrafish

To corroborate the increase in glucose content in *tlr2* mutant zebrafish larvae, an assay using a glucose fluorometric kit was performed. The results showed that glucose content was significantly higher in the *tlr2* mutant than in the wild type larvae, which is consistent with the results obtained by NMR analysis (Figure 6A,B).

We combined the transcriptomic and metabolomic data to explore the potential molecular mechanisms underlying the increase in glucose levels due to *tlr2* deficiency (Figure 7). In Figure 7A, it is shown that the gene transcription levels of most of the enzymes involved in the process of glycolysis, such as *gpib*, *pfkma*, *pgk1*, and *pgam2*, are decreased in the *tlr2* mutant larvae. In the downstream of glycolysis, we found *pck2*, which has a similar function with *pck1*, is decreased (Figure 7B). For the other three significantly enriched signaling pathways associated with metabolism in *tlr2* mutant zebrafish, a decrease in the level of gene transcription of the enzyme is often associated with an increase in the level of its upstream metabolite (Figure 7C,D). For instance, in the valine, leucine, and isoleucine degradation pathway, gene transcript levels of enzymes such as *bcat2* and *echs1* were reduced in the *tlr2* mutant larvae and the levels of the upstream metabolites leucine and valine were elevated (Figure 7D). However, in the tryptophan metabolism pathway, decreased levels of gene transcription for enzymes such as *aldh9a1a.1* and increased levels of the downstream metabolite indole-3-acetate were also found in this pathway (Figure 7C).

## 4. Discussion

TLR2 has been demonstrated to participate in innate immunity as a pro- or anti-inflammation trigger by recognizing ligands from invading pathogens or lysing cells [2,3]. However, whether *tlr2* plays a role in mediating metabolism is still unknown. In this study, we have used zebrafish larvae from a *tlr2* mutant and its wild type siblings to investigate metabolic regulation by Tlr2 under an unchallenged normal developmental condition by combining transcriptomic and metabolomic data at a whole-organism level. The results show that *tlr2* plays a role in regulating metabolism. To further investigate metabolic effects controlled by *tlr2*, we employed NMR spectroscopy and then linked it with the transcriptomic data.

We found that the concentration of glucose was higher in the *tlr2^−/−^* zebrafish larvae compared to the *tlr2^+/+^* (Figure 4, Figure 5 and Figure 6), which was associated with lower expression of several genes involved in the glycolysis and gluconeogenesis pathways (Figure 7). Higher glucose levels in the *tlr2* mutant can result from inhibition of glycolysis and gluconeogenesis. In the present study, we found that *gpib*, *pfkma*, *pgk1*, *pgam2*, and *pck2* genes were all downregulated in the *tlr2* mutant zebrafish larvae (Figure 7). It has been reported that the expression of *Pgk1* is dependent on HIF1A which is induced by the TLR4 and LPS interaction under hypoxia conditions [17], which indicates there might also be a yet undiscovered link between TLR2 and HIF1A regulation. In mammals, the *PCK2* gene encodes the mitochondrial phosphoenolpyruvate carboxykinase 2 (PEPCK-M), which is a gluconeogenic enzyme [44,45]. Thereby, PCK2 has a similar function as PCK1 and is widely expressed by various cell types, whereas PCK1 is mainly expressed in the liver and kidney [45,46,47]. The downregulation of *pck2* in this study could be the result of higher glucose levels in *tlr2* mutant larvae. In addition, various biosynthetic reactions can be fueled by PCK2 activity, which is the reason why the activity of PCK2 also contributes to cell growth and survival during stress [44,48]. In addition to the differences in glucose metabolism in the *tlr2* mutant, we also observed an increase in the levels of various amino acids. Previously we have shown that under wasting conditions, for instance in the absence of the leptin gene, zebrafish larvae contain lower levels of many amino acids [40]. Therefore, the *tlr2* mutant is very different in its metabolic control than the leptin gene.

In a previous study, we found that macrophage and neutrophil migration speed is regulated by *tlr2* upon tissue wounding [28] and mycobacterial infection [29]. Hall et al. reported that mitochondrial metabolism fuel immune cell migration during tissue wounding inflammation [49]. In addition, how metabolism benefits host immune cell defense against invading pathogens has aroused attention. mTOR has been demonstrated to protect host macrophage from mycobacterium-induced cell death through increasing glycolysis-fueled mitochondrial energy [50,51]. Therefore, it could be that the different responses to wounding and infection in the *tlr2* mutant are caused by mitochondrial dysfunction due to the aberrant expression of genes involved in metabolism.

Our RNAseq data suggest that the observed metabolic changes originate intracellularly due to changes in the transcripts of metabolic enzymes. However, mRNA levels do not necessarily predict protein levels, and it is difficult to rule out systemic changes in the mutants. For example, increased levels of branched amino acids in the blood are associated with insulin resistance, which could result in increased glucose. Furthermore, we expect that the observed changes might be the result of transport processes and the interplay of the larval gut with the microbiome. In future experiments we will investigate in which organs the observed changes are most relevant.

Previous evidence for the involvement of TLR signaling in metabolism is based on infectious conditions, whereas we have used unchallenged zebrafish larvae in this study. Therefore, the question remains what is the trigger for activation of TLR2 in these larvae. We have previously shown that in the absence of TLR2 there is an effect on the transcription of many genes involved in neuronal development [27]. It can be speculated that in the absence of TLR2, the apoptotic process in brain development might lead to the production of DAMPs that could trigger TLR2. Alternatively, we can hypothesize that the microbiome controls activation of TLR2. Accumulating evidence suggests that host obesity, insulin resistance, and hormone secretion can be controlled by gut microbes and their derived metabolites [52,53,54,55]. It is generally thought that the interactions between gut microbiota and TLR interactions are responsible for controlling both immunity and metabolism [56]. A previous study from our laboratory shows that *tlr2* is important in regulating microbiota-induced effects on *myd88* transcription [57]. Therefore, we can speculate that the absence of TLR2 could result in dysbiosis of the microbiome. Therefore, the differences of glucose metabolism found in this study may be caused by different compositions of the gut microbiome in *tlr2* mutant and wild type control groups. This could also explain the higher levels of common microbial metabolites such as acetate and indole-3-acetate in the *tlr2* mutant [58]. In future research we want to focus on the role of the microbiome in control of *tlr2* signaling in the presence or absence of infection.

## 5. Conclusions

In this study, we showed that the deficiency of *tlr2* in zebrafish larvae leads to many metabolic changes compared to wild type controls in an unchallenged condition. The metabolic effects of *tlr2* are associated with transcriptional regulation of many enzymes participating in glucose metabolism which could explain the different levels of glucose, lactate, succinate, and malate in *tlr2* mutant zebrafish larvae. In conclusion, *tlr2* has a function in controlling glucose metabolism under normal developmental conditions.

## Figures and Tables

**Figure 1 biology-12-00323-f001:**
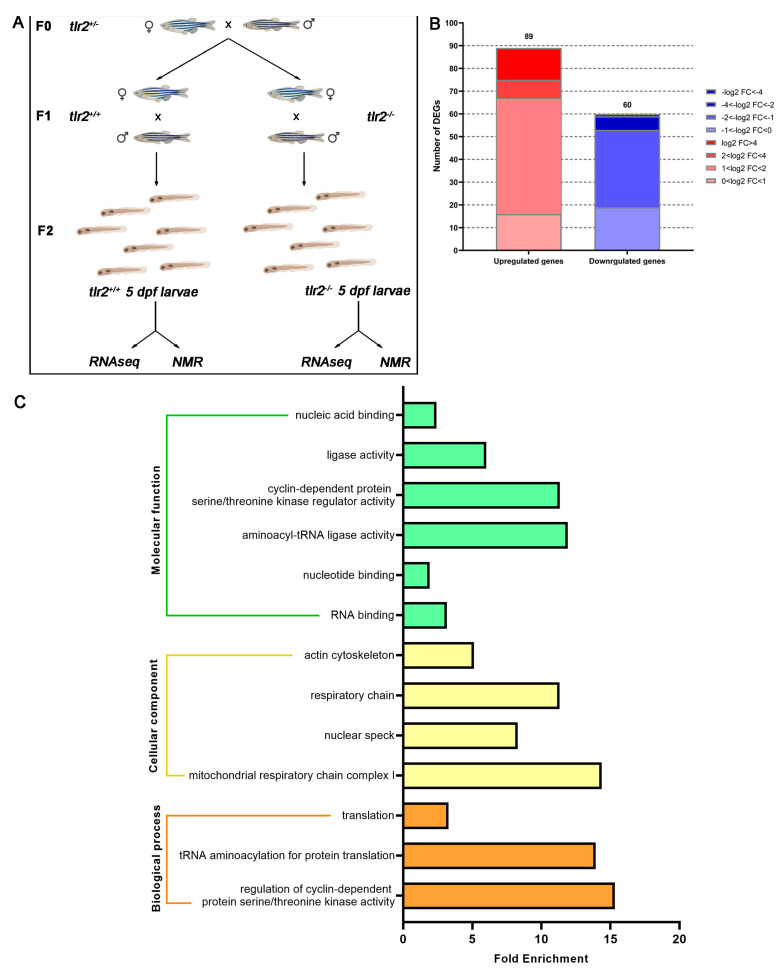
Experimental setup scheme and overview of the RNAseq in *tlr2* mutant zebrafish larvae. (**A**) Setup of the RNAseq and ^1^H NMR experiment. (**B**) Overview of the distribution of differentially expressed genes (DEGs) log2 fold change in *tlr2^−/−^* versus *tlr2^+/+^* zebrafish larvae. DEGs were assessed by *S* value ≤ 0.005. Upregulated gene sets are shown in red, and downregulated gene sets are shown in blue. The intensity of color represents the log2 fold change level. (**C**) Gene ontology (GO) analysis. Significantly enriched GO terms for DEGs of *tlr2^−/−^* versus *tlr2^+/+^* groups were determined by using the hypergeometric test/Fisher’s exact test, with a threshold of *p* value < 0.05 which were adjusted using the Benjamini and Hochbery FDR correction. Genes were categorized according to specific molecular functions, cellular components, and biological processes.

**Figure 2 biology-12-00323-f002:**
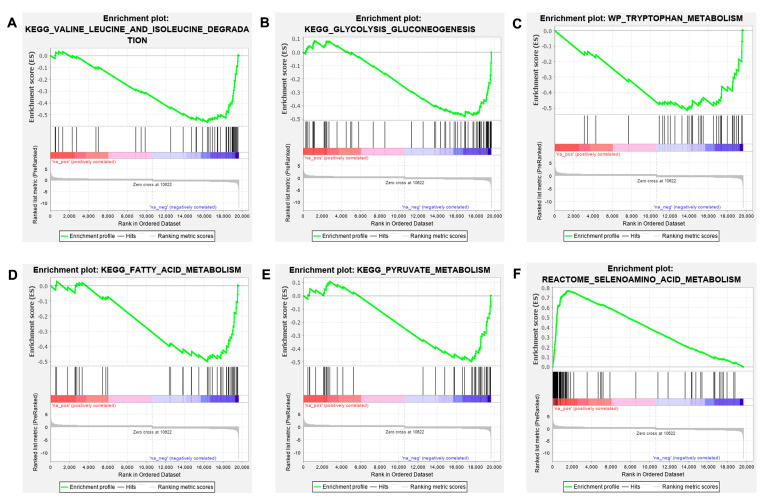
GSEA analysis of the enriched signaling pathways associated with metabolism in *tlr2* wild type and mutant zebrafish larvae. (**A**–**E**) GSEA analysis results for *tlr2* wild type zebrafish larvae group. (**F**) GSEA analysis results for *tlr2* mutant zebrafish larvae group. The gene set C2 (cp.kegg.v.6.2.symbols.gmt) database was used to analyze the whole gene expression dataset of the *tlr2* mutant zebrafish and its wild type control group. For more detailed information see Appendix A.

**Figure 3 biology-12-00323-f003:**
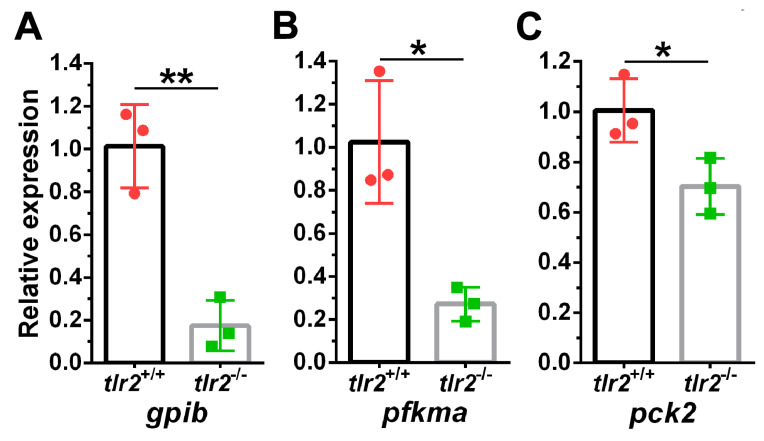
qPCR validation of genes from glycolysis and gluconeogenesis pathway. (**A**) *gpib*. (**B**) *pfkma* (**C**) *pck2*. The information of the genes is shown in Table 1. * represents *p* < 0.05; ** represents *p* < 0.01.

**Figure 4 biology-12-00323-f004:**
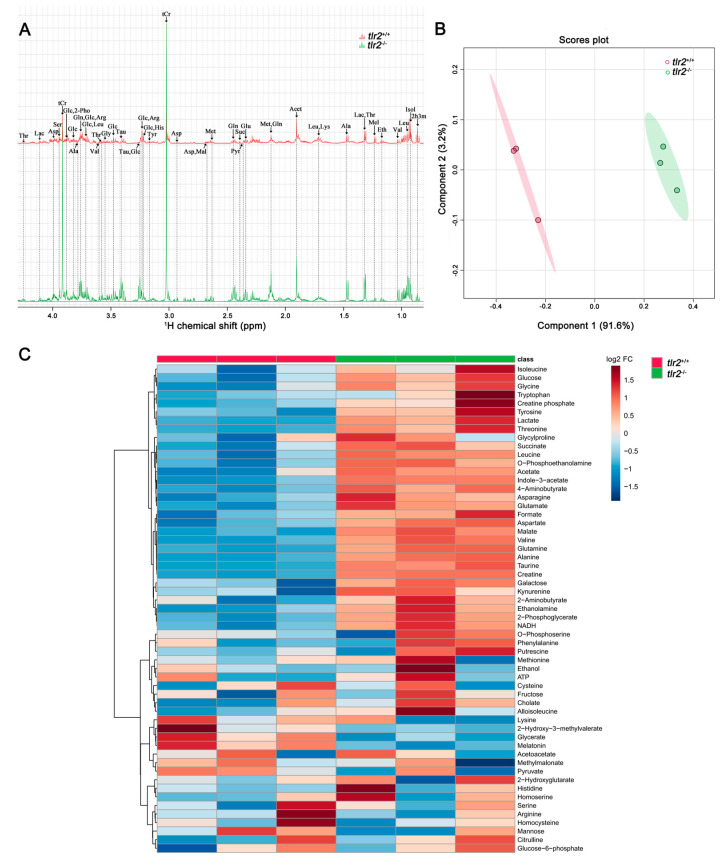
Metabolic profiles of *tlr2* mutant zebrafish larvae measured by NMR and analysis by PLS-DA and heatmap clustering. (**A**) The representative one-dimensional ^1^H NMR spectra of *tlr2* wild type and mutant zebrafish larvae measured by NMR spectrometry. Spectra from chemical shift 0.8 to 4.3 were assigned to specific metabolites. *Thr* threonine, *Lac* lactate, *Asp* aspartate, *Ser* Serine, *tCr* total creatine (creatine + phosphocreatine), *Glc* Glucose, *2-Pho* 2-Phosphoglycerate, *Ala* alanine, *Gln* glutamine, *Arg* arginine, *Leu* leucine, *Val* valine, *Gly* glycine, *Tau* taurine, *His* Histidine, *Tyr* tyrosine, *Mal* Malate, *Met* Methionine, *Suc* Succinate, *Pyr* Pyruvate, *Glu* glutamate, *Acet* Acetate, *Lys* lysine, *Mel* Melatonin, *Eth* Ethanol, *Isol* Isoleucine, *2h3m* 2-Hydroxy-3-methylvalerate. (**B**) PLS-DA analysis of *tlr2* wild type and mutant zebrafish larvae groups. *N* = 3, each replicate represents 120 pooled larvae. (**C**) Heatmap analysis of 57 metabolites detected in *tlr2* wild type and mutant groups. *N* = 3, each replicate represents 120 pooled larvae. FC represents fold change in *tlr2^−/−^* versus *tlr2^+/+^* zebrafish larvae. For more detailed information on significantly differentially expressed metabolites see Appendix A.

**Figure 5 biology-12-00323-f005:**
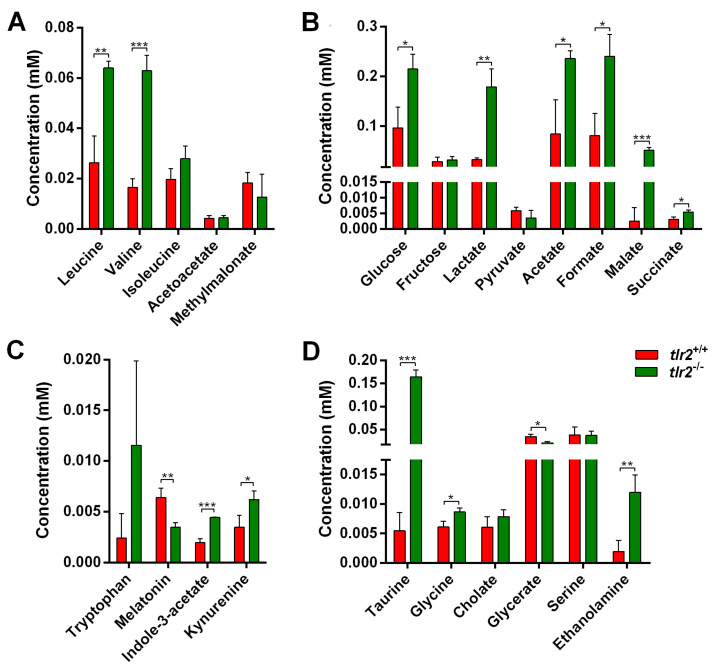
Quantitation of metabolites belonging to four significantly enriched metabolism-related pathways in *tlr2* mutant zebrafish larvae. (**A**) The concentration of the metabolites in the valine, leucine, and isoleucine degradation pathway. (**B**) The concentration of the metabolites in the glycolysis and gluconeogenesis pathway. (**C**) The concentration of the metabolites in the tryptophan metabolism pathway. (**D**) The concentration of the metabolites in the fatty acid metabolism pathway. *N* = 3; * represents *p* < 0.05; ** represents *p* < 0.01; *** represents *p* < 0.001.

**Figure 6 biology-12-00323-f006:**
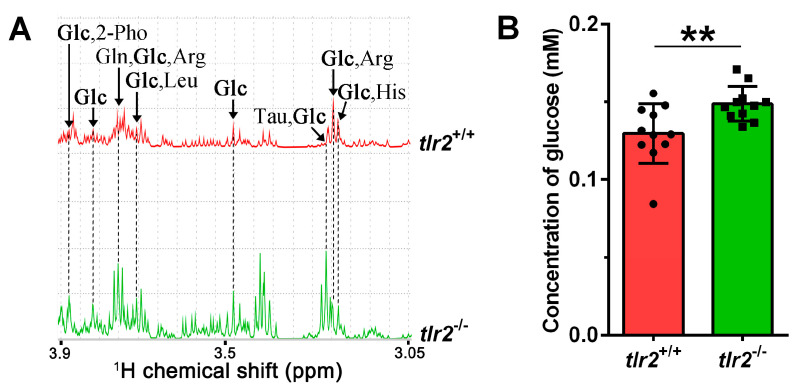
Glucose concentration measurements in zebrafish larvae. (**A**) One-dimensional ^1^H NMR spectra of glucose in *tlr2* wild type and mutant zebrafish larvae. *Glc* Glucose, *2-Pho* 2-Phosphoglycerate, *Gln* glutamine, *Arg* arginine, *Leu* leucine, *Tau* taurine, *His* Histidine. (**B**) Measurement of the glucose concentration from *tlr2* wild type and mutant groups by using glucose kit. Dots and blocks represent individual sample, sample size *N* = 11, each replicate represents five pooled larvae. ** represents *p* < 0.01.

**Figure 7 biology-12-00323-f007:**
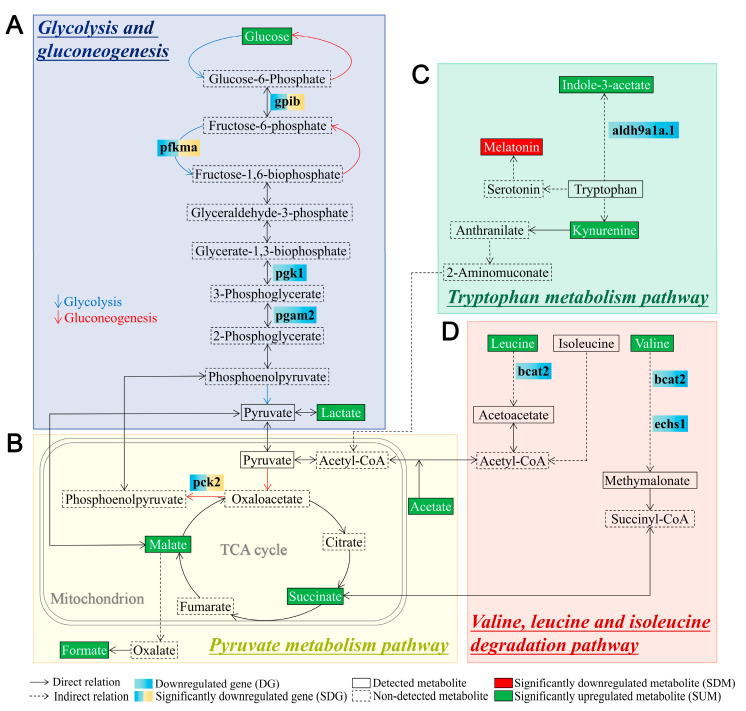
Schematic diagram of differentially expressed metabolites and genes in the significantly enriched signaling pathways associated with metabolism in *tlr2* mutant zebrafish larvae compared to wild type control. (**A**) Glycolysis and gluconeogenesis pathway. (**B**) Pyruvate metabolism pathway. (**C**) Tryptophan metabolism pathway. (**D**) Valine, leucine, and isoleucine degradation pathway. The notes for each icon in the diagram are at the bottom of the diagram. For RNAseq data, setting the *S* value ≤ 0.005 as the cutoff to find significantly expressed genes. For NMR data, significance was established at *p* < 0.05. The genes shown in blue boxes are non-significant downregulated genes in *tlr2* mutant larvae (log2 fold change level and *S* value of *pgk1* is -0.59 and 0.022, *pgam2*: −0.50, 0.061, *aldh9a1a.1*: −0.56, 0.014, *bcat2*: −0.45, 0.089, *echs1*: −0.73, 0.024. Appendix A).

**Table 1 biology-12-00323-t001:** Downregulated genes in glycolysis and gluconeogenesis pathways in *tlr2*^−/−^ versus *tlr2*^+/+^ zebrafish larvae groups.

No.	ENSEMBL ID	Gene Symbol	Gene Name	Log2 Fold Change	*S* Value
1	ENSDARG00000103826	*gpib*	glucose-6-phosphate isomerase b	−1.57	3.24 × 10^−5^
2	ENSDARG00000014179	*pfkma*	phosphofructokinase, muscle a	−1.05	0.00363
3	ENSDARG00000020956	*pck2*	phosphoenolpyruvate carboxykinase 2	−1.80	0.00239

## Data Availability

The RNAseq data are under submission in the NCBI GEO database.

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
