# Peer review of "Transcriptomic and Metabolomic Studies Reveal That Toll-like Receptor 2 Has a Role in Glucose-Related Metabolism in Unchallenged Zebrafish Larvae (Danio rerio)"

_biology, 2023, doi:10.3390/biology12020323_

Round 1
Reviewer 1 Report
This manuscript implies the significance of TLR2 in glucose metabolism in an in vivo model system. However, such a conclusion is too early to state "results demonstrate that Tlr2 has a function in controlling glucose metabolism homeostasis." Glucose metabolism is a potentially complex phenomenon and a plethora of underlying reactions occur during glucose metabolism. Tlr 2 is a known immune regulator and is mostly expressed during viral infections of tumorigenesis.
The major drawback of the study is the methodology and the hypothetical conclusions, i.e. tlr 2 is regulator of glucose metabolism. The first subsection of materials and methods: 2.1 Zebrafish maintenance and samples collection; is very brief, without proper rationale, and no proper selction criteria is described. Just selecting mere tlr 2 mutants and performing RNA seq and NMR doesn't implicate its role in glucose metabolism.
Suggestions:
1. A diabetic-inducing drug like streptazotacin should be induced to fish or similar drug with anti insulin properties.
2. A set of fish (WT and Tlr2 mutant) with drug-induced and non induced, followed by glucose supplementation with proper controls need to be designed.
3. Serum glucose and HBAc1 etc need to be measured along with insulin levels for a period of specific days as per study protocol.
4. Then, you may perform RNA seq and NMR and arrive at a conclusion.
5. If the above experiments are not performed, the paper will be ideally rejected or it may just be converted to a quantitative review: stating implications of tlr2 in plausible glucose and related metabolism using quantitative approaches.
Author Response
This manuscript implies the significance of TLR2 in glucose metabolism in an in vivo model system. However, such a conclusion is too early to state "results demonstrate that Tlr2 has a function in controlling glucose metabolism homeostasis." Glucose metabolism is a potentially complex phenomenon and a plethora of underlying reactions occur during glucose metabolism. Tlr2 is a known immune regulator and is mostly expressed during viral infections of tumorigenesis.
The major drawback of the study is the methodology and the hypothetical conclusions, i.e. tlr2 is regulator of glucose metabolism. The first subsection of materials and methods: 2.1 Zebrafish maintenance and samples collection; is very brief, without proper rationale, and no proper selction criteria is described. Just selecting mere tlr2 mutants and performing RNA seq and NMR doesn't implicate its role in glucose metabolism.
Response:
Glucose metabolism is indeed very complex. That is why we investigated the role of tlr2 under an unchallenged condition. By varying the presence or absence of the tlr2 gene as the only factor, we found that genes and metabolites related to glucose metabolism were different between the two groups, which demonstrated that tlr2 did play a role in glucose metabolism. To be more rigorous, we have changed “has a function” to “plays a role”, see line 20.
2-year-old tlr2 mutants and their wildtype siblings were raised based on the standard protocols (zfin.org). We now mentioned it in more detail in the material and methods see lines 103, 106-107.
Suggestions:
- A diabetic-inducing drug like streptazotacin should be induced to fish or similar drug with anti insulin properties.
- A set of fish (WT and Tlr2 mutant) with drug-induced and non induced, followed by glucose supplementation with proper controls need to be designed.
- Serum glucose and HBAc1 etc need to be measured along with insulin levels for a period of specific days as per study protocol.
- Then, you may perform RNAseq and NMR and arrive at a conclusion.
- If the above experiments are not performed, the paper will be ideally rejected or it may just be converted to a quantitative review: stating implications of tlr2 in plausible glucose and related metabolism using quantitative approaches.
Response:
We really appreciate the suggestions from the reviewer. However, we did not perform these experiments, because the research question was not about diabetes. Although it has been demonstrated that tlr2 as a pro-inflammatory gene is significantly involved in the progression of diabetes, there is no evidence to show tlr2 is a gene that is important for diabetes, We believe the suggested experiments are interesting but fall outside of the scope of our research. In fact, we did measure the glucose level in the entire larvae since there is not sufficient serum in the larvae. The assay we use measures free glucose and does not detect glucose-6-phospahete, therefore we can assume that we measure serum glucose. As far as we know there is no published HBAc1 protocol for zebrafish larvae available and that may not be possible: they are very small and have a low volume.
In this study, we have investigated the role of tlr2 in larval metabolism. It has been demonstrated that many molecular and cellular processes are dependent on glucose metabolism. In our previous paper, we found that tlr2 regulates leukocyte migration upon tissue wounding and mycobacterial infection. In addition, we also found that the tlr2 mutant showed a large difference in the genes involved in glycolysis as compared to heterozygote control zebrafish larvae. This is unexpected since as mentioned by the reviewer its function has been only investigated in inflammation. Based on these reasons, we have further investigated the role of tlr2 in glucose metabolism in this study. We have rewritten the text to make this clear see lines 75-78, and 80-81.
Reviewer 2 Report
The manuscript presents a novel and innovative research. The ideas and findings presented in the paper are original and contribute to the field of study. The research is well-conducted and the methods used are appropriate. This paper is a valuable addition to the literature and has the potential to make a significant impact. However, there might be some minor revisions to improve the language and presentation of the paper:
The discussion in the manuscript is too brief. There are several errors in the format of the reference list and the language used is not consistent. Revisions are necessary to improve the overall quality of the paper. Specifically, the discussion should be expanded to provide more in-depth analysis and the references should be checked for accuracy and proper formatting. Additionally, the language should be polished to ensure clarity and adherence to academic conventions.
Author Response
The manuscript presents a novel and innovative research. The ideas and findings presented in the paper are original and contribute to the field of study. The research is well-conducted and the methods used are appropriate. This paper is a valuable addition to the literature and has the potential to make a significant impact. However, there might be some minor revisions to improve the language and presentation of the paper:
The discussion in the manuscript is too brief. There are several errors in the format of the reference list and the language used is not consistent. Revisions are necessary to improve the overall quality of the paper. Specifically, the discussion should be expanded to provide more in-depth analysis and the references should be checked for accuracy and proper formatting. Additionally, the language should be polished to ensure clarity and adherence to academic conventions.
Response:
We have checked all the references in the manuscript and added more references to the discussion. Furthermore, we have extended the discussion see lines 379-384 and 387-394.
Reviewer 3 Report
The manuscript by Hu et al from the Spaink lab, ”Transcriptomic and metabolomic studies reveal that Toll-like 2 Receptor 2 controls metabolism in unchallenged zebrafish larvae (Danio rerio)”, explores the role of tlr2 in metabolism in the absence of infection. By comparing the transcriptome and metabolome of tlr2+/+ and tlr2-/- larvae, the authors demonstrated congruent changes in the mRNA of several metabolic enzymes and their substrates. Specifically, the mutants had an increase of glucose, lactate, succinate, malate, and branched amino acids. The data are of good quality. The paper is well written. The discussion is thoughtful. One aspect that I found unsatisfactory pertains to the interpretation of the results. Although the authors are probably correct to suggest that all the changes originate intracellularly due to the observed changes of the transcripts of their metabolic enzymes, mRNA levels do not necessarily predict protein levels, and it is difficult to rule out systemic changes in the mutants. For example, increased blood branched amino acids are associated with insulin resistance, which could result in increased glucose.
Author Response
The manuscript by Hu et al from the Spaink lab, ”Transcriptomic and metabolomic studies reveal that Toll-like 2 Receptor 2 controls metabolism in unchallenged zebrafish larvae (Danio rerio)”, explores the role of tlr2 in metabolism in the absence of infection. By comparing the transcriptome and metabolome of tlr2+/+ and tlr2-/- larvae, the authors demonstrated congruent changes in the mRNA of several metabolic enzymes and their substrates. Specifically, the mutants had an increase of glucose, lactate, succinate, malate, and branched amino acids. The data are of good quality. The paper is well written. The discussion is thoughtful. One aspect that I found unsatisfactory pertains to the interpretation of the results. Although the authors are probably correct to suggest that all the changes originate intracellularly due to the observed changes of the transcripts of their metabolic enzymes, mRNA levels do not necessarily predict protein levels, and it is difficult to rule out systemic changes in the mutants. For example, increased blood branched amino acids are associated with insulin resistance, which could result in increased glucose.
Response:
We completely agree with the reviewer and have added a sentence in the discussion to make this clear. We have taken the liberty of using the formulation of the reviewer since we really like this statement.
Our RNAseq data suggest that the observed metabolic changes originate intracellularly due to changes of the transcripts of metabolic enzymes. However, mRNA levels do not necessarily predict protein levels, and it is difficult to rule out systemic changes in the mutants. For example, increased blood branched amino acids are associated with insulin resistance, which could result in increased glucose. Furthermore, we expect that the observed changes might be the result of transport processes and the interplay of the larval gut with the microbiome. In future experiments we will investigate in which organs the observed changes are most relevant. See lines 387-394.
Round 2
Reviewer 1 Report
The authors have somewhat addressed the suggestions. The following revisions are needed for better clarity of manuscript:
1. Revise the title as "Transcriptomic and metabolomic studies reveal that Toll-like 2 Receptor 2 has a role in glucose-related metabolism in unchallenged zebrafish (Danio rerio).
2. How can a sentence in the Abstract begin from "Except for their function.. ". Revise it to as "Toll-like receptors (TLRs) have been implicated in the regulation of various metabolism pathways, besides being involved in innate immunity"
3. The Introduction should clarify more on the homology of zebrafish and human TLRs and why zebrafish could be used as an analogous human model as the entire manuscript is targeted towards human health development.
4. The Discussion should narrate how Tlr 2 could also modulate glucose metabolism in humans, which genes it could affect and under different clinical scenario.
5. Some recent references on the same could be cited such as https://doi.org/10.3389/fbioe.2021.721717, https://doi.org/10.1016/j.scitotenv.2021.147989 ,
Author Response
The authors have somewhat addressed the suggestions. The following revisions are needed for better clarity of manuscript:
1. Revise the title as "Transcriptomic and metabolomic studies reveal that Toll-like 2 Receptor 2 has a role in glucose-related metabolism in unchallenged zebrafish (Danio rerio).
Response:
We have changed the title into “Transcriptomic and metabolomic studies reveal that Toll-like 2 Receptor 2 has a role in glucose-related metabolism in unchallenged zebrafish (Danio rerio).”
2.How can a sentence in the Abstract begin from "Except for their function.. ". Revise it to as "Toll-like receptors (TLRs) have been implicated in the regulation of various metabolism pathways, besides being involved in innate immunity"
Response:
We have changed the sentence into “Toll-like receptors (TLRs) have been implicated in the regulation of various metabolism pathways, in addition to its function in innate immunity”. See lines: 22-23.
3. The Introduction should clarify more on the homology of zebrafish and human TLRs and why zebrafish could be used as an analogous human model as the entire manuscript is targeted towards human health development.
Response:
We have mentioned the homology of zebrafish and human TLRs, see lines 75-76. In addition, we further described the similar function between mammalian TLR2 and zebrafish tlr2, see lines 76-78. We would like to mention that we have already published two extensive reviews on the function of TLR receptors in vertebrates including zebrafish so we don’t want to repeat the background information. I hope the referee can check our reviews (PMID: 35205112; PMID: 24560981) and please let us know if more references are needed.
4. The Discussion should narrate how Tlr 2 could also modulate glucose metabolism in humans, which genes it could affect and under different clinical scenario.
Response:
We have already discussed how tlr2 could modulate glucose metabolism in the discussion part, and clarified the importance of pgk1 and pck2 genes for glucose metabolism regulation in the absence of the tlr2 gene. see lines 362-381. We would like to stress that there is hardly anything known about the function of TLRs in embryonic metabolism so since this novelty of our paper there are no other things we can discuss. Please note that embryonic metabolism is in no way comparable to metabolism in diabetic animals.
5. Some recent references on the same could be cited such as https://doi.org/10.3389/fbioe.2021.721717, https://doi.org/10.1016/j.scitotenv.2021.147989,
Response:
We don’t see a connection with the suggested references with toll-like receptors, perhaps the referee can check whether the correct link is supplied.
Reviewer 2 Report
I agree to receive the revised article based on the author's modifications.
Author Response
I agree to receive the revised article based on the author's modifications.
Response: We received this comment.